# Facile Preparation of iPP Fibrous Membranes from In Situ Microfibrillar Composites for Oil/Water Separation

**DOI:** 10.3390/polym17152114

**Published:** 2025-07-31

**Authors:** Chengtao Gao, Li Zhang, Xianrong Liu, Chen He, Shanshan Luo, Qin Tian

**Affiliations:** 1Guizhou Material Industrial Technology Institute, Guiyang 550014, China; 2College of Electronic Information Engineering, Guiyang University, Guiyang 550005, China; 18386253396@163.com; 3College of Materials Science and Engineering, Guiyang University, Guiyang 550005, China

**Keywords:** oily wastewater, oil/water separation, oil recovery, isotactic polypropylene, microfiber composites

## Abstract

Superhydrophobic and superoleophilic nanofibrous or microfibrous membranes are regarded as ideal oil/water separation materials owing to their controllable porosity, superior separation efficiency, and ease of operation. However, developing efficient, scalable, and environmentally friendly strategies for fabricating such membranes remains a significant challenge. In this study, isotactic polypropylene (iPP) fibrous membranes with morphologies ranging from ellipsoidal stacking to microfiber stacking were successfully fabricated via a multistage stretching extrusion and leaching process using in situ microfibrillar composites (MFCs). The results establish a significant relationship between microfiber morphology and membrane oil adsorption performance. Compared with membranes formed from high-aspect-ratio microfibers, those comprising short microfibers feature larger pores and a more open structure, which enhances their oil adsorption capacity. Among the fabricated membranes, the iPP membrane with an ellipsoidal stacking morphology exhibits optimal performance, achieving a porosity of 65% and demonstrating both hydrophobicity and superoleophilicity, with a silicone oil adsorption capacity of up to 312.5%. Furthermore, this membrane shows excellent reusability and stability over ten adsorption–desorption cycles using chloroform. This study presents a novel approach leveraging in situ microfibrillar composites to prepare high-performance oil/water separation membranes in this study, underscoring their considerable promise for practical use.

## 1. Introduction

Oil pollution resulting from oil spills or industrial oily wastewater has now become an urgent global environmental problem, so the purification of oily wastewater has attracted much attention worldwide [1]. To date, various approaches have been developed to conquer this global issue, including combustion, filtration, and collection [2,3]. Among these methods, filtration and collection via absorbents, with no secondary pollution generated and the collected oil being reusable, have attracted intensive interest. Conventional adsorption materials, including activated carbon, zeolites, and wool fibers, are limited by inherent shortcomings such as low separation efficiency and inadequate regeneration capability, which restrict their practical applications [4,5,6]. Consequently, the development of novel, straightforward, and efficient adsorption materials for oil-contaminated water remediation remains challenging and imperative.

It is widely acknowledged that an ideal adsorption or separation material must possess two critical characteristics: (1) hydrophobic–oleophilic or hydrophilic–oleophobic surface properties to ensure oil/water selectivity and anti-fouling performance; (2) a porous structure to achieve high separation efficiency [7,8,9]. Based on these criteria, researchers have developed a variety of novel porous materials (e.g., metal meshes, aerogels, foams, sponges, and membrane materials [10,11,12,13,14]) for oil/water separation. Among these, fibrous membranes are recognized as promising materials owing to their three-dimensionally interconnected porous architecture, elevated specific surface area, superior separation efficiency, high flux, and ease of operation [15,16]. Currently, electrospinning is the primary method for fabricating fibrous membranes [17,18,19,20]. By adjusting process parameters or incorporating inorganic/organic fillers, fibrous membranes with excellent oil/water separation performance can be prepared [21,22,23,24]. For example, Yufan Deng et al. conducted an innovative study to fabricate stereocomplex polylactic acid (PLA) porous fibers using electrospinning technology. The fiber membrane exhibits excellent lipophilicity, robust mechanical properties, and high hydrolysis resistance, which collectively contribute to its superior oil adsorption capacity. Specifically, the maximum oil adsorption capacities reached 148.9 g/g at 23 °C and 114.8 g/g at 60 °C [24]. However, this technique is limited by its complex manufacturing process, poor reproducibility, low production capacity, and environmental concerns, which hinder its commercial application [25,26,27,28]. Therefore, developing an efficient, scalable, and environmentally friendly strategy for fibrous membrane fabrication remains an urgent research priority.

In situ microfiber composites (MFCs), a polymer blending material composed of microfibril bundles formed in situ during shear-melt extrusion or hot/cold stretching, have been widely used to prepare reinforced polymer blends or novel barrier products [29,30,31]. In our previous studies on MFCs, we observed that microfibers with high aspect ratios interconnect and form membranes after etching away the matrix phase, suggesting that the MFC concept could offer an environmentally friendly approach for the continuous production of fibrous membranes [32,33,34]. However, the application potential of fibrous membranes derived from MFCs for oily wastewater treatment remains unexplored in the published literature.

This study presents a novel strategy for fibrous membrane fabrication leveraging in situ microfibrillar composites (MFCs). As a prominent commercial semicrystalline polymer, isotactic polypropylene (iPP) boasts a range of exceptional properties, including outstanding chemical and moisture resistance, robust mechanical performance, versatile processability, and cost-effectiveness. These attributes collectively endow iPP with significant potential for application in the field of oil/water separation. Hence, in this work, novel isotactic polypropylene (iPP) fibrous membranes featuring a three-dimensional interpenetrating pore network were fabricated through multistage stretching extrusion and selective etching techniques. This method overcomes the limitations of traditional electrospinning, producing fibrous membranes demonstrating excellent hydrophobicity, superoleophilicity, and a high silicone oil adsorption capacity (up to 312.5%), the material also exhibits robust reusability and stability. This strategy opens up new avenues for the preparation of efficient and environmentally friendly materials for oil/water separation. Notably, other polymer materials, such as polyethylene (PE), PLA, PVDF, polyamide, and polyester, can also be easily transformed into fibrous membranes with three-dimensional interconnected channels using this method.

## 2. Experimental

### 2.1. Materials

Commercial isotactic polypropylene (iPP) granules (grade T30S), serving as the microfibrillar phase, were sourced from Sinopec Yangzi Petrochemical Company Ltd. (Nanjing, Jiangsu Province, China). This iPP grade exhibits a melt flow index (MFI) of 3.0 g/10 min (230 °C, 2.16 kg) and a density of 0.90 g/cm^3^. Four grades of ethylene-vinyl acetate copolymer (EVA) with varying vinyl acetate (VA) contents were utilized as the matrix phase. These grades, sourced from Tosoh Corporation (Yokkaichi, Japan), include EVA 625 (15% VA content), EVA 630 (16%), EVA 633 (20%), and EVA 634 (26%). From EVA 625 to EVA 634, the viscosity ratio between EVA and iPP increases gradually, as reported in our previous work [35]. All chemical reagents (xylene, cyclohexane, Sudan III, chloroform, ethanol) were obtained from Rong He Wei Chuang Co. (Guiyang, China) as commercially available products and used without further purification.

### 2.2. Preparation of iPP Fibrous Membrane

Prior to the preparation of the isotactic polypropylene (iPP) fibrous membranes, both iPP and ethylene-vinyl acetate (EVA) were oven-dried for 24 h at 80 °C and 40 °C, respectively. Subsequently, in situ microfibrillar composites (MFCs), comprising iPP as the microfibrillar phase and EVA as the matrix, were fabricated via multi-stage stretching extrusion. As illustrated in Figure 1, EVA/iPP pellets with a constant mass ratio of 85:15 (*w/w*) were extruded from a twin-screw extruder, connector, and laminating–multiplying elements (LMEs). The iPP phase gradually elongated into microfibrils by shearing the flow field during processing, and finally formed an in situ microfibrillar composite that was 1.0 mm thick and 200 mm wide. The temperature profile of the twin-screw extruder was set at 100 °C, 190 °C, 200 °C, and 200 °C, respectively. The temperature of the LMEs stayed at 200 °C. The screw speed was 153 r/min. Rectangular specimens sectioned from as-extruded EVA/iPP MFCs were encased in copper mesh and subsequently immersed in hot xylene (115 °C) for 5 h. This process selectively etched away the EVA matrix, yielding an iPP fibrous membrane. For conciseness, the iPP membranes derived from EVA 625/iPP, EVA 630/iPP, EVA 633/iPP, and EVA 634/iPP MFCs are designated as iPP-1, iPP-2, iPP-3, and iPP-4, respectively.

### 2.3. Scanning Electron Microscopy (SEM)

The morphology of isotactic polypropylene (iPP) fibrous membranes was characterized using a Quanta 250 FEG scanning electron microscope (SEM) (FEI Co., Ltd., Hillsboro, OR, USA) at 10 kV accelerating voltage. SEM images were recorded after the surfaces of the samples were coated with gold.

### 2.4. Porosity

The ethanol saturation method (ESM) was applied to explore the porosity of the iPP fibrous membranes [36]. A piece of membrane (0.01 g) was immersed in 100 mL of ethanol for 6 h; after 6 h, the membrane was taken out and weighed (*w*_0_). Then, the membrane was put into a vacuum oven to remove the absorbed ethanol. The membrane weight (*w*) was measured following ethanol desorption. Porosity (*P*) was then calculated using Equation (1):(1)$$P=\frac{(w_{0}-w)\rho_{m}}{\rho_{m}w_{0}+\left(\rho_{m}-\rho_{e}\right)w}\times 100\%$$
where *ρ_m_* (0.9 g∙cm^−3^) and *ρ_e_* (0.8 g∙cm^−3^) are the density of iPP and ethanol, respectively. The initial mass (*w*_0_) of the membrane is determined after 6 h of immersion in 100 mL of ethanol. *w* is the weight of the membrane after releasing the absorbed ethanol into a vacuum oven.

### 2.5. Contact Angle Measurements

The hydrophobic and lipophilic characteristics of the isotactic polypropylene (iPP) fibrous membrane were assessed using an OCA 20 contact angle goniometer (DataPhysics Instruments, Raiffeisenstraße 34, 70794 Filderstadt, Germany) with water and cyclohexane as probe liquids at room temperature (RT). The average contact angle was obtained from five measurements.

### 2.6. Oil/Water Separation Performance Test

Chloroform and cyclohexane were employed as heavy and light oil, respectively. In the oil–water separation test, both probe oils were stained with Sudan Ⅲ. The chloroform and cyclohexane, with a larger and lower density compared to water, sunk under the water or floated on the water, respectively. Fibrous membrane segments, immobilized with tweezers, were employed to adsorb the oil samples.

### 2.7. Oil Adsorption Tests

To assess the oil absorption capacity (static) of the iPP fibrous membranes towards various oils, membrane specimens were initially submerged in vessels containing 100 mL of oil for 3 h. After 3 h, the membrane was taken out with tweezers and held for several seconds until no more oil dropped down; then, the weight of the membrane after absorbing the oil was carefully weighed. The saturated oil adsorption capacity (*Q*) was calculated based on Equation (2) [37,38]:(2)$$Q(\%)=\frac{w_{2}-w_{1}}{w_{1}}\times 100$$
where *w*_1_ and *w*_2_ represent the initial and final mass of the fibrous membrane. Various organic solvents and oils were employed to assess the oil sorption capacity of the iPP fibrous membranes derived from in situ microfibrillar composites. After weighing the *w*_2_, all membranes were put into a vacuum oven to release chloroform. Then, the membrane was immersed into chloroform to absorb chloroform again. To evaluate the recyclability and recoverability of the fibrous membranes for oil cleanup applications, adsorption/desorption cycles were conducted over 10 consecutive repetitions.

## 3. Results and Discussion

### 3.1. The Morphology of the iPP Fibrous Membrane

It is well documented that the microfibrillar morphology in MFCs is mainly controlled by the viscosity ratio; compatibility; interfacial tension between the matrix and dispersed phase; concentration; and processing parameters including the temperature, draw ratio, intensity of the shear flow field, etc. [39,40,41]. In this study, EVA/iPP MFCs with different microfibrillar morphologies were fabricated by tailoring the viscosity ratio between the matrix and microfibrillar phase using four grades of EVA (Figure 1), and Figure 1 exhibits the microfibrillar morphologies of the iPP fibrous membranes after etching the EVA matrix away. It can be clearly observed that the iPP microfibrillar morphology and the diameter of the iPP microfibrils strongly depend on the viscosity ratio, as previously reported [35]. In the fabrication of the EVA/iPP MFC with the lowest viscosity ratio, the iPP particles only deformed into ellipsoids (Figure 1a). With the increase in viscosity ratio, iPP microfibrils with a larger aspect ratio formed accordingly; the diameter of iPP microfibrils decreased from 1.55 µm to 0.33 µm, as displayed in our previous work [35]. The deformation mechanism of iPP in EVA/iPP MFCs with different viscosity ratios was investigated in our previous work, which will not be discussed in this study. As shown in Figure 1, the iPP ellipsoids or microfibrils with different diameters are randomly distributed and interconnected, forming a fibrous membrane with a 3D porous structure. On the contrast, the iPP ellipsoids or microfibrils separate from each other and orient along the extrusion direction in the as-extruded samples, which is comparable with other MFCs. This phenomenon is ascribed to the fact that undissolved iPP microfibrils completely rearrange with the aid of solvent during the etching process [35]. Furthermore, these ellipsoids’ or microfibrils’ stacking morphologies in iPP fibrous membranes also form lots of interconnected pores, which is quite similar to the structure of electrospun fibrous membranes, where different sizes can be clearly found. However, the porous architecture within each fibrous membrane is predominantly governed by the inherent morphology of the constituent microfibers. To be specific, the porous structure in membranes stacked with short microfibrils is looser than that consisting of long microfibrils, which might contribute to the difference in the degree of entanglement. Nonetheless, iPP fibrous membranes with inherent hydrophobicity and a controllable 3D interpenetrated porous structure are successfully obtained from MFCs. Based on these findings, iPP fibrous membranes exhibit excellent oil absorption properties and demonstrate significant potential as an effective material for oil/water separation applications.

### 3.2. Surface Wettability and Porosity of iPP Fibrous Membranes

For an excellent oil absorbent, the primary requirement necessitates that the material surface exhibits both hydrophobicity and oleophilicity [22,42]. Contact angle measurements of distilled water and cyclohexane were employed to assess the surface wettability of microfibrous cellulose (MFC)-derived isotactic polypropylene (iPP) membranes (Figure 2). Representative images of methylene blue-dyed water droplets and Sudan III-stained cyclohexane droplets on the membrane surfaces are also presented. The results demonstrate that the microfibrillar morphology of isotactic polypropylene (iPP) fibrous membranes exerts a strong influence on their water contact angle.

The iPP fibrous membrane with an ellipsoidal morphology (iPP-1) possesses the largest water contact angle, up to 134°. With the elongation of the iPP microfibrillar phase, the shape of water droplets on the iPP fibrous membrane surface transforms from spherical into hemispherical, resulting in a continuous decrease in the water contact angle [42]. The result is primarily attributable to the different microfibrillar morphology in each iPP fibrous membrane. In addition, according to Figure 1, iPP fibrous membranes with long microfibrils have a greater degree of phase entanglement than those with ellipsoidal bodies, which results in the formation of smaller pores and a denser porous structure. Generally, larger pores and a looser porous structure can trap much more air, decreasing the contact area between water droplets and the membrane surface [43]. Therefore, iPP-1 consisting of ellipsoids possesses excellent hydrophobicity with the largest water contact angle. Meanwhile, all iPP fibrous membranes exhibit superoleophilicity, as evidenced by the instantaneous spreading of cyclohexane oil droplets under capillary action, resulting in contact angles of 0°. No significant changes can be observed in the oil contact angle for each iPP fibrous membrane from Figure 2. However, the recorded videos during the testing process demonstrate that the permeation speed of oil into the membranes slightly decreases from iPP-1 to iPP-4, indicating that the porous structure becomes denser with the formation of long microfibrils. The results demonstrate that the iPP fibrous membrane with an ellipsoidal stacking morphology presents better hydrophobic and superoleophilic surface wettability compared to the other membranes, which endows it an excellent oil adsorption capacity.

In addition to surface wettability, the porosity of the porous structure is another important parameter for achieving a highly efficient adsorbent. As we know, high porosity provides more space and enhanced driving force for oil uptake. Figure 3 illustrates the porosity of the iPP fibrous membranes, measured using the ethanol saturation method (ESM). All the iPP fibrous membranes obtained from MFCs exhibit high porosity (larger than 60%), which is comparable with other porous oil absorbents [35]. As expected, iPP-1, consisting of short microfibrils, shows the highest porosity of up to 65%. With the elongation of the iPP microfibrils, they exhibit a marginal reduction in porosity, attributable to increased inter-fibrillar entanglement and the development of a compacted porous network.

### 3.3. Oil/Water Separation Performance

The above results demonstrate that iPP fibrous membranes obtained from MFCs possess excellent hydrophobicity, superoleophilicity, and high porosity, which are conducive to the application of iPP fibrous membranes in oil/water separation. The excellent oil/water selectivity, high oil adsorption capacity, and good reusability of iPP fibrous membranes demonstrate that fibrous membranes prepared from in situ microfibrillar composites are also promising candidates for oily-water treatment, which is similar to electrospun fibrous membranes. Consequently, the representative iPP-1 membrane, exhibiting the highest water contact angle and porosity among the series, was selected to assess oil/water separation performance. Figure 4 illustrates the separation process of immiscible oils from an oil/water emulsion using this membrane. For visual differentiation, the probe oils (cyclohexane and chloroform) were stained red with Sudan III. Because of the density difference between oils and water, the stained cyclohexane droplet floats on the water, while the chloroform droplet sinks under the water. As shown in Figure 4, once the iPP-1 fibrous membrane gets close to either the cyclohexane on the water surface or the underwater chloroform, the probe oils undergo immediate absorption, resulting in the complete removal of dye contaminants and yielding purified water devoid of residual oil-soluble dye. The superior oil/water separation capability of iPP fibrous membranes can be ascribed to two aspects. First, the hydrophobic/superoleophilic surface makes the membrane selectively absorb oil and repel water. Second, the high porosity of the 3D interconnected channel provides an enhanced capillary force to take up oils.

The adsorption capacity of an oil absorbent plays an important role in determining its practical oil/water separation application. To further investigate this performance, a piece of iPP fibrous membrane was immersed into oil for 3 h, and its saturated oil adsorption capacity (*Q*) was determined based on the weight before and after absorbing oils. Figure 5a presents the *Q* of the iPP fibrous membranes towards different oils/organic solvents. As shown in Figure 5a, it can be clearly seen that the iPP fibrous membranes can absorb a wide range of oils/organic solvents and attain high adsorption capacities in the range of 146% to 479% according to the density and viscosity of the oils. Moreover, the adsorption ability of the iPP fibrous membranes is controlled by the microfibrillar morphology. The iPP-1 membrane presents the highest oil adsorption capacities for almost all probe oils compared with other iPP fibrous membranes containing long microfibrils, which is attributed to the synergistic effect of its 3D interconnected pore channels, high porosity, and superoleophilicity. However, the adsorption capacity (*Q*) is currently determined using immersion-based saturation tests in a static setup, which may overestimate performance relative to real-world continuous flow or percolation conditions. Therefore, future work should focus on evaluating the adsorption capacity under turbulent conditions to provide a more accurate representation of practical performance.

The reusability of an absorbent is another key parameter for practical oil cleanup applications. The iPP fibrous membrane sample was repeatedly immersed in chloroform for adsorption and then put into a vacuum oven for desorption. Figure 5b displays the reusability of the iPP fibrous membranes in the adsorption application for chloroform. No significant deterioration in oil adsorption capacity was observed for iPP-1, iPP-2, and iPP-3 during the first seven adsorption/desorption cycles. After seven adsorption/desorption cycles, the adsorption capacity of the iPP fibrous membranes gradually decreased with the number of cycles, which might be ascribed to the fact that the porous structure in the membrane is partially blocked by the adsorbed oils. Nonetheless, the iPP-1 fibrous membrane still exhibited high oil adsorption capacity up to 392.1% after 10 cycles, showing an outstanding recyclability for dealing with oil/water separation.

The excellent oil/water selectivity, high oil adsorption capacity, and good reusability of iPP fibrous membranes demonstrate that fibrous membranes prepared from in situ microfibrillar composites are also promising candidates for oily-water treatment, which is similar to fibrous membranes fabricated by electrospinning (as shown in Table 1). Meanwhile, lots of technologies can be used to prepare MFCs (including common extrusion with hot or cold drawing [44], shear flow extrusion techniques like slit extrusion [45], multistage stretching extrusion [46], triangle-arrayed triple-screw extruders, etc.). Moreover, a wide range of polymers can be transformed into microfibrils under shear flow or the drawing process and, thus, into fibrous membranes via MFCs. Nevertheless, the fabrication of fibrous membranes using the MFC strategy necessitates a solvent etching step. The underlying mechanisms by which the solvent etching process influences the performance of the resulting oil/water separation membranes remain to be fully elucidated and warrant further investigation.

## 4. Conclusions

In summary, the novel approach of fabricating isotactic polypropylene (iPP) fibrous membranes via in situ microfibrillar composites (MFCs) demonstrates significant promise for oily-wastewater treatment. This potential is evidenced by their excellent oil/water selectivity, high oil adsorption capacity, and favorable reusability. The membranes exhibit excellent hydrophobicity (water contact angle: 134°) and superoleophilicity (cyclohexane contact angle: 0°), underpinning their high selectivity. Similarly to electrospun membranes, the MFC-derived iPP membranes possess a 3D interconnected porous structure, enabling ultrahigh adsorption capacities for diverse oils and organic solvents, with a maximum capacity of 479% for chloroform. However, the oil adsorption performance is strongly dependent on the microfibrillar morphology within the MFCs. Specifically, shorter microfibrils promote the formation of membranes with larger pore sizes and a looser porous network compared to longer fibrils with higher aspect ratios, leading to enhanced adsorption capacity. Furthermore, the iPP fibrous membranes exhibit good reusability, which is comparable with other oil adsorbents. The MFC concept offers versatility beyond iPP, as it is applicable to various polymers (e.g., polyolefins, polyesters, polyamides) in constructing tailored fibrous membranes. Consequently, this study demonstrates the significant potential of the MFC strategy for the commercial-scale fabrication of high-performance fibrous membranes for oil/water separation applications. However, further investigations on promoting the stability of MFC-derived fibrous membranes under extreme environmental conditions and the separation of stabilized emulsions should be undertaken.

## Data Availability

The data that support the findings of this study are available from the corresponding author, LSS and TQ, upon reasonable request.

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
