# Peer review of "Facile Preparation of iPP Fibrous Membranes from In Situ Microfibrillar Composites for Oil/Water Separation"

_polymers, 2025, doi:10.3390/polym17152114_

Round 1

Reviewer 1 Report

Comments and Suggestions for Authors

The manuscript is focus on the Facile preparation of iPP fibrous membranes from in situ microfibrillar composites for oil/water separation. The manuscript is interesting, however I have few queries before accepting its for the publication. 

  1. Why authors choose isotactic polypropylene.
  2. Is there any changes in the fiber diameter in ipp-1, 2, 3, and 4. please check and estalished relation between size and oil water separation efficiency. 
  3. Why porosity is almost same???
  4. Authors should discuss about mechanism... 
  5. Authors should incorporate comparative table with other electrospiiing fibers.

Author Response

We must thank all reviewers for the critical feedback. We feel lucky that our manuscript went to these reviewers as the valuable comments from them not only helped us with the improvement of our manuscript, but suggested some neat ideas for future studies.

Based on the comments we received, careful modifications have been made to the manuscript. All changes were marked in red text. We hope the new manuscript will meet the magazine’s standard.

1. Why authors choose isotactic polypropylene.

As one of commercial semicrystalline polymers, isotactic polypropylene (PP) possesses several extraordinary properties involving good mechanical properties, excellent chemical and moisture resistance, versatile processability and low cost,  which has been widely applied in various fields including oil/water separation.   

2. Is there any changes in the fiber diameter in iPP-1, 2, 3, and 4. please check and estalished relation between size and oil water separation efficiency.

The fiber diameters in iPP-1, 2, 3, and 4 were previously reported in our earlier study (Journal of Applied Polymer Science 2019, 136 (21), 47557) and are not reiterated in the current research. We appreciate the reviewer's insightful suggestion. In response, we have now included more detailed information on fiber morphology, including descriptions of fiber diameters, in the revised manuscript.  

 Diameter distributions of iPP fibers (a) iPP-1; (b) iPP-2; (c) iPP-3; (d) iPP-4.

3. Why porosity is almost same.

 The porosity of fibrous membranes is influenced by both the morphology of the fibers and the leaching process. Our study revealed that the morphology of iPP microfibrils has only a minor impact on porosity. In the future, we will investigate the effects of the leaching process on the porosity and oil/water separation performance of fibrous membranes.

4. Authors should discuss about mechanism.

Thanks for the reviewer's valuable suggestion! More detailed mechanism descriptions have been added in the revised manuscript.

Reviewer 2 Report

Comments and Suggestions for Authors

The draft of the article discusses the urgent issue of creating materials with varying wettability to separate water-oil mixtures. The authors propose a novel approach, the creation of microfibrillar composites, to develop such materials. Although the manuscript provides new information in this area, it contains some flaws and inaccuracies and should be recommended for publication after corrections.

1) Introduction:

There are no specific examples, although there are many articles in the scientific literature on the study of membranes based on PP).

The authors need to justify the choice of PP.

There is an error in the last paragraph: the authors write that the material can absorb 312.5 g/g. But in fact, these are percentages and will only be 3.125 g/g. In accordance with these results, I would like to note that the material is not outstanding in its properties. The authors need to compare the material they have developed with analogs described in the literature; perhaps your membranes are not competitive (This comparison must be presented, preferably at the end of the discussion).

2) Point 2.2

Present the characteristics of EVA. Why is the EVA/PP ratio 85/15? How will the properties change if the ratio is changed? According to your technology, 85 parts of EVA are irretrievably lost?

Scheme 1(a) – no captions to the numbers on the extruder.

3) Morphology.

In the SEM images presented by the authors, no differences in morphology are visible. However, the authors refer to their previous work, and the differences are clearly visible there.

4) Point 3.2

Remove percentages in the discussion, use g/g.

Here the authors talk about excellent separation ability and high sorption capacity, but without comparison - this is unfounded and unsubstantiated. In addition, the purity of water and oil after separation should be assessed.

5) Conclusions.

Conclusions should be written more specifically based on the results obtained.

"enabling ultrahigh adsorption capacities for diverse oils and organic solvents, reaching a maximum of 479% for chloroform."

"Furthermore, the iPP fibrous membranes exhibit good reusability, comparable to established oil sorbents."

The authors make unsubstantiated conclusions.

Author Response

1. Introduction:

There are no specific examples, although there are many articles in the scientific literature on the study of membranes based on PP.

The authors need to justify the choice of PP. 

There is an error in the last paragraph: the authors write that the material can absorb 312.5 g/g. But in fact, these are percentages and will only be 3.125 g/g. In accordance with these results, I would like to note that the material is not outstanding in its properties. The authors need to compare the material they have developed with analogs described in the literature; perhaps your membranes are not competitive (This comparison must be presented, preferably at the end of the discussion.

In the revised manuscript, we have incorporated specific examples, provided a thorough justification for the selection of isotactic polypropylene (iPP), corrected the identified errors, and added a comparison of the oil adsorption capacities of fibrous membranes prepared by electrospinning and those made from MFCs, all in accordance with the reviewer's suggestions.

2. Point 2.2

Present the characteristics of EVA. Why is the EVA/PP ratio 85/15? How will the properties change if the ratio is changed? According to your technology, 85 parts of EVA are irretrievably lost?

Scheme 1(a)-no captions to the numbers on the extruder..

The characteristics of EVA and captions to the numbers on the extruder have been included in the revised manuscript according to the reviewer’s suggestion. The microfibrillar morphology in multilayered fiber composites (MFCs) is primarily governed by several key factors, including the viscosity ratio, compatibility, interfacial tension between the matrix and dispersed phases, concentration, and processing parameters such as temperature, draw ratio, and the intensity of the shear flow field. In this study, EVA/iPP MFCs with distinct microfibrillar morphologies were fabricated by adjusting the viscosity ratio between the matrix and microfibrillar phases using four different grades of EVA. Specifically, the ratio of EVA to iPP was maintained at a constant 85/15.

3. Morphology.

In the SEM images presented by the authors, no differences in morphology are visible. However, the authors refer to their previous work, and the differences are clearly visible there.

 The SEM images are related to the selected regions. The samples in this study is the same as that in the previous study (Journal of Applied Polymer Science 2019, 136 (21), 47557), but different regions were chosen for the SEM analysis.

4. Point 3.2

Remove percentages in the discussion, use g/g.

Here the authors talk about excellent separation ability and high sorption capacity, but without comparison - this is unfounded and unsubstantiated. In addition, the purity of water and oil after separation should be assessed.

The adsorption capacity can be expressed as a percentage (For example: Applied Materials Today, 2017, 9:77-81), the unit of adsorption capacity has been unified to percentage, according to the reviewer’s suggestion.

The revised manuscript now includes a comparison of the oil adsorption capacities of fibrous membranes prepared by electrospinning and those made from MFCs.

The contact angle results revealed that the iPP fibrous membrane with an ellipsoidal stacking morphology exhibited hydrophobic and superoleophilic surface wettability. The water droplet on the iPP fibrous membrane surface maintained a spherical shape, indicating high selectivity of the membrane during oil/water separation. Consequently, we did not analyze the purity of the water and oil after separation.

5. Conclusions.

Conclusions should be written more specifically based on the results obtained.

"enabling ultrahigh adsorption capacities for diverse oils and organic solvents, reaching a maximum of 479% for chloroform."

"Furthermore, the iPP fibrous membranes exhibit good reusability, comparable to established oil sorbents."

The authors make unsubstantiated conclusions.

 According to the reviewer's valuable suggestion, conclusions have been revised.

Reviewer 3 Report

Comments and Suggestions for Authors

General Summary

This manuscript presents an innovative and environmentally friendly approach for producing isotactic polypropylene (iPP) fibrous membranes using in situ microfibrillar composites (MFCs) for oil/water separation. The research is relevant and the results are promising regarding oil adsorption efficiency, porosity, hydrophobicity, and reusability. The methodology is clear, and the data are well-organized and discussed. However, some methodological aspects and result interpretations require deeper clarification, especially regarding extrusion control, thermal characterization of the blends, and quantitative relationships between morphology, porosity, and performance. Practical applications could also be expanded to consider more realistic oil/water separation scenarios.

General suggestions:

1- Include more quantitative correlations between structure and performance to explain, based on numerical or graphical data, how the structural properties of membranes (morphology, porosity, fiber aspect ratio, etc.) are directly related to their functional performance, especially in oil adsorption capacity or oil/water separation efficiency;

2- Discuss limitations of the MFCs-based strategy, addressing potential weaknesses, challenges or technical constraints of the MFCs manufacturing strategy, especially in comparison to other conventional approaches;

3- If possible, add thermal analysis data to evaluate phase compatibility;

4- Expand the discussion on applicability in real situations (e.g. in an oil spill response, or in emulsified systems, pH/salinity effects).

Specific Comments:

Section 2.1, page 2

"This iPP grade exhibits a melt flow index (MFI) of 3.0 g/10 min (230 °C, 2.16 kg)"

- Please clarify the rationale for using iPP with an MFI of 3.0 g/10 min. Mid-range MFI values can lead to challenges in controlling the fiber morphology during multistage stretching. It would be helpful to include a brief discussion of how the polymer's flow properties may have influenced fiber dimensions and uniformity.

Section 2.2, page 2

"Four grades of ethylene-vinyl acetate copolymer (EVA) (Tosoh Corporation, Yokkaichi, Japan; grades: EVA 625, EVA 630, EVA 633, EVA 634) were employed as the matrix phase. From EVA 625 to EVA 634, the viscosity ratio between EVA and iPP increases gradually, as reported in our previous work."

-  Given the complexity of iPP/EVA blends, if possible, I recommend including DSC or TGA thermal analysis to verify phase compatibility, which would support the interpretation of the microfibrillar morphologies. In addition, I ask that the authors cite and add in the references which work they reported the information.

Section 2.2, page 3

“Rectangular specimens sectioned from as-extruded EVA/iPP MFCs were encased in copper mesh and subsequently immersed in hot xylene (115 °C) for 5 hours. This process selectively etched away the EVA matrix, yielding an iPP fibrous membrane.”

- Please provide more detail regarding the multistage stretching conditions and the control parameters used prior to xylene immersion. Given the importance of fiber morphology, clarifying the stretching ratio, stretching speed and temperature profile, as a way to contribute to methodological reproducibility.

Section 3.1, page 4

“As shown in Figure 1, the iPP ellipsoids or microfibrils are randomly distributed and interconnected, forming a fibrous membrane with a 3D porous structure.”

- The authors could consider presenting quantitative correlations between the aspect ratio of the microfibrils, the membrane’s porosity, and the oil adsorption capacity (Q%). Including a plot or regression analysis would add rigor to the interpretation of structure–performance relationships.

Section 3.3, page 7

"Figure 4 illustrates the separation process of immiscible oils from an oil–water emulsion using this membrane."

- The authors describe the separation of immiscible oils (cyclohexane and chloroform) from an oil–water emulsion. However, it is not clear whether the emulsions were stabilized (e.g., surfactant-based) or how representative they are of real oily wastewater. To enhance practical relevance, consider discussing membrane performance under more complex scenarios, such as stable emulsions, saline water, or variable pH conditions, which are common in marine or industrial settings.

Section 3.3, page 8

"To further investigate this performance, a piece of iPP fibrous membrane was immersed into oil for 3h, and its saturated oil adsorption capacity (Q) was determined based on the weight before and after absorbing oils."

- Please clarify whether the adsorption capacity (Q) was measured in a static or dynamic setup. Immersion-based saturation tests may overestimate performance compared to real-world continuous flow or percolation conditions. Including complementary tests under flow-through or agitation conditions would enhance the practical relevance of the Q data. The authors can at least suggest future tests in turbulent conditions.

Section 3.3, page 9

“Figure 5(a) presents the Q of iPP fibrous membranes towards different oils/organic solvents.” and “Figure 5(b) displays the reusability in the adsorption application of iPP fibrous membrane towards chloroform.”

- Figure 5 presents valuable comparative data; however, it would enhance the scientific contribution to explore statistical correlations between fiber morphology (e.g., aspect ratio, surface roughness) and the oil adsorption capacities (Q%).

Were regression analyses or correlation coefficients evaluated across different oils?

Furthermore, regarding Figure 5(b), the authors could discuss the potential fouling or degradation mechanisms responsible for the decline in Q over the cycles.

Could SEM or surface chemistry analyses after multiple runs support the hypothesis of pore blockage by adsorbed oils?

Section 4, page 10

The conclusion effectively summarizes the main results, emphasizing the membrane’s adsorption performance, selectivity, and reusability. However, the authors could briefly mention some potential limitations to guide future developments, such as: structural stability under varying pH, temperature, or surfactants; environmental concerns regarding the use of xylene and chloroform; and the membrane’s performance in stabilized emulsions. It would also be helpful to indicate which fabrication steps may pose challenges for industrial scaling.

Recommendation

The manuscript presents a relevant contribution to the field of oil and water separation materials. The approach using microfibrillar composites is technically sound and environmentally promising. I recommend minor revisions, with the suggestions above intended to improve the clarity of the methodology and expand the discussion on practical applications.

Comments on the Quality of English Language

The manuscript demonstrates originality and technical relevance. The methodology based on microfibrillar composites is solid and aligns well with the journal's scope. There are no concerns regarding self-citations, and the work has potential to contribute meaningfully to both academic and applied research after appropriate revisions. If necessary, I will continue to be available to evaluate the manuscript after revisions.

Author Response

General suggestions:

1. Include more quantitative correlations between structure and performance to explain, based on numerical or graphical data, how the structural properties of membranes (morphology, porosity, fiber aspect ratio, etc.) are directly related to their functional performance, especially in oil adsorption capacity or oil/water separation efficiency.

The fibers are intertwined with each other, making it difficult to accurately determine the aspect ratio. Therefore, it is difficult to establish a quantitative relationship between structure and performance.

2. Discuss limitations of the MFCs-based strategy, addressing potential weaknesses, challenges or technical constraints of the MFCs manufacturing strategy, especially in comparison to other conventional approaches.

 In the revised manuscript, we have thoroughly addressed the limitations of the MFCs-based strategy according to the reviewer's valuable suggestion.

3. If possible, add thermal analysis data to evaluate phase compatibility.

 The phase compatibility between iPP and EVA has been thoroughly investigated in our previous work (Polymer International, 2023, 72(6): 597-603) and will not be reiterated in this study.

4. Expand the discussion on applicability in real situations (e.g. in an oil spill response, or in emulsified systems, pH/salinity effects).

In the revised manuscript, the discussion section has been expanded to address the applicability in real-world scenarios according to the reviewer's valuable suggestion. 

Specific Comments:

1.Section 2.1, page 2

"This iPP grade exhibits a melt flow index (MFI) of 3.0 g/10 min (230 oC, 2.16 kg)"

Please clarify the rationale for using iPP with an MFI of 3.0 g/10 min. Mid-range MFI values can lead to challenges in controlling the fiber morphology during multistage stretching. It would be helpful to include a brief discussion of how the polymer's flow properties may have influenced fiber dimensions and uniformity.

The microfibrillar morphology in multilayered fiber composites (MFCs) is primarily governed by several key factors, including the viscosity ratio, compatibility, interfacial tension between the matrix and dispersed phases, concentration, and processing parameters such as temperature, draw ratio, and the intensity of the shear flow field. In this study, EVA/iPP MFCs with distinct microfibrillar morphologies were fabricated by adjusting the viscosity ratio between the matrix and microfibrillar phases using four different grades of EVA.

2. Section 2.2, page 2

"Four grades of ethylene-vinyl acetate copolymer (EVA) (Tosoh Corporation, Yokkaichi, Japan; grades: EVA 625, EVA 630, EVA 633, EVA 634) were employed as the matrix phase. From EVA 625 to EVA 634, the viscosity ratio between EVA and iPP increases gradually, as reported in our previous work."

Given the complexity of iPP/EVA blends, if possible, I recommend including DSC or TGA thermal analysis to verify phase compatibility, which would support the interpretation of the microfibrillar morphologies. In addition, I ask that the authors cite and add in the references which work they reported the information.

The phase compatibility between iPP and EVA has been thoroughly investigated in our previous work (Polymer International, 2023, 72(6): 597-603) and will not be reiterated in this study.

The reference (our previous work) has been added in the revised manuscript.

3. Section 2.2, page 3

“Rectangular specimens sectioned from as-extruded EVA/iPP MFCs were encased in copper mesh and subsequently immersed in hot xylene (115 °C) for 5 hours. This process selectively etched away the EVA matrix, yielding an iPP fibrous membrane.”

Please provide more detail regarding the multistage stretching conditions and the control parameters used prior to xylene immersion. Given the importance of fiber morphology, clarifying the stretching ratio, stretching speed and temperature profile, as a way to contribute to methodological reproducibility.

The multistage stretching conditions and the control parameters have been added in the revised manuscript according to the reviewer's valuable suggestion.

4. Section 3.1, page 4

“As shown in Figure 1, the iPP ellipsoids or microfibrils are randomly distributed and interconnected, forming a fibrous membrane with a 3D porous structure.”

The authors could consider presenting quantitative correlations between the aspect ratio of the microfibrils, the membrane’s porosity, and the oil adsorption capacity (Q%). Including a plot or regression analysis would add rigor to the interpretation of structure–performance relationships.

The fibers are intertwined with each other, making it difficult to accurately determine the aspect ratio. Therefore, it is not possible to establish a quantitative relationship between the aspect ratio of the microfibrils, the membrane’s porosity, and the oil adsorption capacity (Q%).

5. Section 3.3, page 7

"Figure 4 illustrates the separation process of immiscible oils from an oil–water emulsion using this membrane."

The authors describe the separation of immiscible oils (cyclohexane and chloroform) from an oil–water emulsion. However, it is not clear whether the emulsions were stabilized (e.g., surfactant-based) or how representative they are of real oily wastewater. To enhance practical relevance, consider discussing membrane performance under more complex scenarios, such as stable emulsions, saline water, or variable pH conditions, which are common in marine or industrial settings.

We appreciate the reviewer's valuable suggestions! In our future work, we will investigate the oil/water separation performance of fibrous membranes derived from microfibrillar composites (MFCs) under more complex conditions.

6. Section 3.3, page 8

"To further investigate this performance, a piece of iPP fibrous membrane was immersed into oil for 3h, and its saturated oil adsorption capacity (Q) was determined based on the weight before and after absorbing oils."

Please clarify whether the adsorption capacity (Q) was measured in a static or dynamic setup. Immersion-based saturation tests may overestimate performance compared to real-world continuous flow or percolation conditions. Including complementary tests under flow-through or agitation conditions would enhance the practical relevance of the Q data. The authors can at least suggest future tests in turbulent conditions.

The adsorption capacity (Q) was measured in a static setup, as detailed in the revised manuscript. Future work will focus on investigating the adsorption capacity (Q) under turbulent conditions.

7. Section 3.3, page 9

“Figure 5(a) presents the Q of iPP fibrous membranes towards different oils/organic solvents.” and “Figure 5(b) displays the reusability in the adsorption application of iPP fibrous membrane towards chloroform.”

Figure 5 presents valuable comparative data; however, it would enhance the scientific contribution to explore statistical correlations between fiber morphology (e.g., aspect ratio, surface roughness) and the oil adsorption capacities (Q%).

Were regression analyses or correlation coefficients evaluated across different oils?

Furthermore, regarding Figure 5(b), the authors could discuss the potential fouling or degradation mechanisms responsible for the decline in Q over the cycles.

Could SEM or surface chemistry analyses after multiple runs support the hypothesis of pore blockage by adsorbed oils?

The aspect ratio and surface roughness of fibers are difficult to accurately determine.

No, regression analyses or correlation coefficients were not evaluated across different oils in this study.

SEM and surface chemistry analyses are unable to observe the changes in the internal pore structure after multiple cycles. Therefore, they cannot directly support the hypothesis of pore blockage by adsorbed oils.

8. Section 4, page 10

The conclusion effectively summarizes the main results, emphasizing the membrane’s adsorption performance, selectivity, and reusability. However, the authors could briefly mention some potential limitations to guide future developments, such as: structural stability under varying pH, temperature, or surfactants; environmental concerns regarding the use of xylene and chloroform; and the membrane’s performance in stabilized emulsions. It would also be helpful to indicate which fabrication steps may pose challenges for industrial scaling.

The conclusion has been revised according to the reviewer’s suggestion.

Round 2

Reviewer 1 Report

Comments and Suggestions for Authors

Accept

Author Response

We must thank all reviewers for the critical feedback. We feel lucky that our manuscript went to these reviewers as the valuable comments from them not only helped us with the improvement of our manuscript, but suggested some neat ideas for future studies.

Based on the comments we received, careful modifications have been made to the manuscript. All changes were marked in red text. We hope the new manuscript will meet the magazine’s standard.

Reviewer 2 Report

Comments and Suggestions for Authors

Dear authors, you really did a great job and made corrections on every point. Unfortunately, there are points with which I categorically disagree. Please note that at the last stage of the review I suggested you to use g/g (instead of %) as a unit of measurement of oil absorption. And you had an error in the introduction, where you indicated an overestimated sorption capacity for your materials (indicated g/g instead of percent). After making changes, you decided to leave the percent (okay, it is not a critical remark), but in Table 1 you provide a comparison of your materials with those presented in the scientific literature and here we cannot agree with you. I checked references 47-50 and they all use g/g as the unit of measurement, which you put in table 1 without converting to percentages (1 g/g = 100%). It feels like you are deliberately trying to make your results stand out, but they are not.

Provide a fair comparison of your samples.

Author Response

We must thank all reviewers for the critical feedback. We feel lucky that our manuscript went to these reviewers as the valuable comments from them not only helped us with the improvement of our manuscript, but suggested some neat ideas for future studies.

Based on the comments we received, careful modifications have been made to the manuscript. All changes were marked in red text. We hope the new manuscript will meet the magazine’s standard.

Responding to Reviewer #2:

  1. Dear authors, you really did a great job and made corrections on every point. Unfortunately, there are points with which I categorically disagree. Please note that at the last stage of the review I suggested you to use g/g (instead of %) as a unit of measurement of oil absorption. And you had an error in the introduction, where you indicated an overestimated sorption capacity for your materials (indicated g/g instead of percent). After making changes, you decided to leave the percent (okay, it is not a critical remark), but in Table 1 you provide a comparison of your materials with those presented in the scientific literature and here we cannot agree with you. I checked references 47-50 and they all use g/g as the unit of measurement, which you put in table 1 without converting to percentages (1 g/g = 100%). It feels like you are deliberately trying to make your results stand out, but they are not.

Provide a fair comparison of your samples..

In the revised manuscript, Table 1 has been corrected in accordance with the reviewer's suggestions. The value of Q in Reference 48 remains unchanged, as it was originally presented in percentages (as illustrated below). 

Oil/water separation test

The capability of the SupREME fibers to absorb oils, including hexane, heptane, octyl aldehyde, petroleum ether, and olive oil, was examined by measuring the quantity of the oil absorbed in the fibrous matrices. The oil absorbency was calculated using the equation.(Sorry, the formula cannot be pasted here. Please forgive me.)

Round 3

Reviewer 2 Report

Comments and Suggestions for Authors

I agree with the authors' corrections. In this version, the article can be recommended for publication.